# OpenReview forum: "Agree to Disagree? A Meta-Evaluation of LLM Misgendering"
_colmweb.org/COLM/2025/Conference — COLM 2025_

### Official Review · Reviewer_Zan3 · 2025-05-06

**Rating:** 7
**Confidence:** 3
**Ethics Flag:** 1

**Summary:**

This paper presents an in-depth analysis of evaluation strategies for measuring LLM misgendering. In particular, it addresses the problem by converting datasets used for probabilistic evaluations into those for generative assessment and vice versa. These parallel datasets allow for both probabilistic and generative evaluations, enabling a comparison of outcomes to achieve more stable and robust results. When evaluating multiple LLMs on these datasets, the findings indicate that the two evaluation strategies did not agree in approximately 20% of the instances, confirming the limitations of a single approach and the absence of a valid and reliable evaluation protocol. The paper concludes with a list of recommendations for evaluating LLM misgendering.

The paper addresses an interesting, novel, and relevant topic for the NLP community and confirms the lack of a proper methodology to evaluate LLM misgendering. This type of paper is quite important because it clearly demonstrates the limitations of the current methodologies and recommends crucial directions for future work in the field.

The structure of the paper is straightforward and clear. It is generally easy to follow. The experimental part is dense, with several tables and figures, but this is necessary to support the claims.

The human evaluation is an important component of the paper that allows readers to support and appreciate the results of the automatic evaluation. All the findings would lose their meaning without human evaluation. A potential issue is that the human evaluation was conducted by two authors of the paper, which could introduce bias. A better approach would be to have independent evaluators to ensure the fairness of the analysis.

**Questions To Authors:**

Although it is written at the end of the reproducibility statement, it is fundamental that the authors share the data to foster research on this topic.

**Reasons To Accept:**

The topic addressed in the paper is relevant for the NLP community.

 The generation of parallel datasets for probabilistic and generative evaluations is novel and quite effective in studying and assessing the LLM misgendering problems from multiple angles.
The experimental part is solid, and human evaluation is an important contribution.

The final recommendations are an important plus for fostering research on this topic and helping the LLM’s users.

**Reasons To Reject:**

The human evaluation that is a core part of the paper has been performed by the authors of the paper, who might have some biases due to their involvement in the paper.

The evaluation part is dense with a lot of results and discussion.

---

> ### Author Response · Authors · 2025-06-02
> **Response to Reviewer Zan3**
>
> Thanks for your helpful feedback! To address each of your comments:
>
> > “A potential issue is that the human evaluation was conducted by two authors of the paper, which could introduce bias.”, “the authors [...] might have some biases due to their involvement in the paper.”
>
> Unlike for other tasks in NLP (e.g., toxicity detection, summarization), annotating model generations for correct pronoun usage is **rarely subjective**. Moreover, as we mention in Line 231, both the authors are **experts** in English-language pronoun usage. Additionally, the annotators did not look at the generation-based evaluation results while annotating, thereby **minimizing confirmation bias**.
>
> In contrast to correct pronoun usage, annotating for other categories of misgendering (e.g., pronoun avoidance, extraneous gendered language) *can* be subjective. However, the annotators determined these categories inductively through a **pilot annotation and discussion** (see Appendix G), bringing in their knowledge of trans people and gendering.
>
> -----
>
> > “The experimental part is dense, with several tables and figures, but this is necessary to support the claims.”, “The evaluation part is dense with a lot of results and discussion.”
>
> We would be happy to consider any suggestions to reduce the density of results.
>
> -----
>
> > “it is fundamental that the authors share the data to foster research on this topic.”
>
> We will share our data in the camera-ready version of our paper.
>
> -----
>
> > “confirms the lack of a proper methodology to evaluate LLM misgendering”
>
> We would like to clarify that we do not argue that the probability- and generation-based evaluation methodologies are inherently improper, but rather that a probability-based evaluation of misgendering can be inappropriate for a generation-based downstream task, and vice versa. Hence, practitioners should carefully consider the alignment of their evaluation methodology with their downstream use case (see Section 7).

---

> > ### Comment · Reviewer_Zan3 · 2025-06-05
> > **ACK**
> >
> > Thanks a lot for answering my questions and doubts. Your answers confirm my initial impression that this is a good paper.

---

### Official Review · Reviewer_U3xB · 2025-05-10

**Rating:** 7
**Confidence:** 2
**Ethics Flag:** 1

**Summary:**

The paper compares probabilistic (cloze-style) and generative evaluation methods used to measure LLM misgendering. They present a formalization to extend probabilistic (generative) datasets to (generative) counterparts and measure correlation at the dataset level for 6 different models. Specifically, this is implemented by a) generating text pre-/post- [MASK] tokens in the probabilistic setting and checking the accuracy of the first generated pronoun with the ground truth and b) by creating a template from the generative dataset (text is truncated to a single pronoun) and the pronoun is replaced with [MASK] token. To validate these automatic metrics and gain qualitative insights, the authors conduct human evaluations on the generative versions of two misgendering datasets (MISGENDERED and RUFF).

**Questions To Authors:**

1) Is there any explanation on why the correlations are higher for MISGENDERED than the TANGO dataset?
2) Was there any human evaluation on whether the truncating of text for extending generative to probabilistic evaluation doesn't necessarily loose the context required for having a single pronoun as the gold standard?

**Reasons To Accept:**

1. The study is conducted across several datasets and language models, providing complementary insights on extending the generative and probabilistic evaluations.

2. The analysis from human evaluations are insightful in that they provide additional signals on how misgendering can take several forms beyond the usage of pronouns. Specifically the examples provided in the appendix are very helpful.

**Reasons To Reject:**

The paper presents a valuable comparison between misgendering rates in open-ended generation and probabilistic evaluations. However, the authors’ interpretation of divergence between these two evaluation types as evidence of limited ecological validity may conflate distinct dimensions of model behavior. Open-ended generation reflects a model’s prior tendencies and biases in unconstrained language production. In contrast, cloze-style tasks probe a model’s ability to interpret and complete structured prompts, often under tightly controlled conditions. The lack of correlation between the two is therefore not inherently problematic; rather, it highlights the different capacities being measured.  A model can recognize correct pronouns yet still generate biased outputs and vice versa.

The paper’s motivation, as stated in Line 47, rests on the observation that NLP work often fails to clarify which dimension is being measured, yet no citation is provided to support this claim.

Additionally, in Lines 88–94, the relationship between "trick tests" and either evaluation paradigm remains ambiguous. If "trick tests" are meant to correspond to cloze-style evaluations or even prompts for open-ended generations, I would question the equivalence: is there a reason to believe this equivalence?

The recommendations are general and lack actionable guidance. For example, the authors do not clarify whether probabilistic evaluations are in fact used as proxies for open-ended misgendering in practice, nor do they analyze the specific consequences of such mismatches. also, while the paper touches on important considerations on the contextual nature of misgendering, it does not offer the human validation in their own study (see Q2).

---

> ### Author Response · Authors · 2025-06-02
> **Response to reviewer U3xB**
>
> Thanks for your detailed and thoughtful feedback!
>
> > “The authors’ interpretation of divergence between these two evaluation types as evidence of limited ecological validity may conflate distinct dimensions of model behavior. [...] The lack of correlation between the two is therefore not inherently problematic”
>
> We **completely agree** with this. In L42-47, we emphasize that the validity of probability and generation-based evaluations is contextual, but papers are not always explicit about how their evaluation setup aligns with the downstream task (L45-57). We argue that the divergence between probability and generation-based evaluation results is not inherently problematic but becomes problematic when: (1) cloze-style evaluations are used to measure misgendering in open-ended generation tasks (e.g., chat-based applications), and (2) generation-based evaluations are used to evaluate misgendering in, e.g., sequence ranking for automatic speech recognition.
> In sum, both evaluation methodologies have their time and place, but their divergence reflects that they are not substitutes for each other. We can include the following intuitive clarification in our camera-ready version: “A model may understand misgendering but not generate it, and vice versa, and this is precisely why it is important to have aligned and valid evaluations.”
>
> -----
>
> > “The paper’s motivation, as stated in Line 47, rests on the observation that NLP work often fails to clarify which dimension is being measured, yet no citation is provided to support this claim.”, “the authors do not clarify whether probabilistic evaluations are in fact used as proxies for open-ended misgendering in practice.”
>
> Thanks for pointing this out! Our missing citations here were an oversight, and **we will include the following citations** to support our claim.
>
> For NLP bias work more broadly, we will cite [1], which found that “bias tests are often poorly reported, missing critical details about what the paper conceptualises as the bias or harm to be measured, and sometimes even details about how the test is constructed.”
>
> In the context of misgendering, [2] claims in the abstract that “when an individual is introduced with a pronoun, models can mostly faithfully reuse this pronoun in the next sentence.” [3] “evaluates popular language models’ ability to use declared third-person personal pronouns.” These excerpts suggest that the authors of [2] and [3] may see applications of their work to generation contexts as well; however, the main bodies of [2] and [3] only consider cloze-style evaluations.
>
> -----
>
> > “the relationship between ‘trick tests’ and either evaluation paradigm remains ambiguous.”
>
> “Trick tests” do not correspond directly to cloze-style or generation-based evaluations. In [4], an example trick test is “{occupation} is [MASK]”. In this setting, the probabilities of masculine vs. feminine tokens filling the [MASK] are measured. Functionally, “trick tests” are similar to probability-based measurements of bias, i.e., they consider candidate tokens that could fill [MASK] from controlled sets. However, “trick tests” are intrinsic measurements of bias, in contrast to the probability-based evaluations we consider in our paper, which assess the ability of models to fill the [MASK] given richer context with a ground truth pronoun and a single correct answer. We cite [4] in our related works as the authors, like us, encourage LLM practitioners to reliably measure bias with downstream tasks in mind.
>
> -----
>
> > “nor do they analyze the specific consequences of such mismatches.”
>
> The consequences of these mismatches are the **instance, dataset, and model-levels of disagreement in evaluation results that we document in Section 5**, raising questions about the usability of measurements made with a given evaluation methodology. For example, if the probability-based version of MISGENDERED ranks model A better than model B while the generation-based version of MISGENDERED ranks model B better than model A, how can we choose between A and B for deployment without more knowledge about the downstream task? Invalid measurements can lead to poor model selection, faulty model deployments, or public misinformation about model performance, causing real harm to trans individuals.
>
> -----
>
> [1] [This prompt is measuring <mask>: evaluating bias evaluation in language models](https://aclanthology.org/2023.findings-acl.139/) (Goldfarb-Tarrant et al., Findings 2023)
>
> [2] [Robust Pronoun Fidelity with English LLMs: Are they Reasoning, Repeating, or Just Biased?](https://aclanthology.org/2024.tacl-1.95/) (Gautam et al., TACL 2024)
>
> [3] [MISGENDERED: Limits of Large Language Models in Understanding Pronouns](https://aclanthology.org/2023.acl-long.293/) (Hossain et al., ACL 2023)
>
> [4] [Bias in language models: Beyond trick tests and toward ruted evaluation](https://arxiv.org/abs/2402.12649) (Lum et al., arXiv 2024)

---

> > ### Author Response · Authors · 2025-06-02
> > **Response to reviewer U3xB's questions**
> >
> > > “Q: Is there any explanation on why the correlations are higher for MISGENDERED than the TANGO dataset?”
> >
> > Tables 1 and 2 suggest that the correlations are higher for TANGO than MISGENDERED. This is due to the constrained nature of [MASK]-filling. In the probability-based version of MISGENDERED, we only allow pronouns to fill the [MASK]. However, in the pre-[MASK] generation-based version of MISGENDERED, any token can be generated in the former position of the [MASK] (i.e., does not need to be a pronoun) and a pronoun can be generated at any point in the sequence.
> >
> > In contrast, in the generation-based version of TANGO, a pronoun can be generated at any point in the sequence. However, in the probability-based version of TANGO, we construct templates from model generations, and therefore, we only allow pronouns (i.e., not any token) to fill the [MASK] in the position where the model had already generated a pronoun. Since autoregressive decoding is intended to be a high-quality sampler for the actual distributions over sequences encoded by LLMs, this likely explains the higher correlations.
> >
> > -----
> >
> > > “Q: Was there any human evaluation on whether the truncating of text for extending generative to probabilistic evaluation doesn't necessarily loose the context required for having a single pronoun as the gold standard?”
> >
> > Our constructed templates contain the context from the generation, e.g., “Casey is an American actor and they are known for their roles in film,” which always establishes a subject and ground-truth pronoun. The generations are then truncated directly before the second pronoun in the completion, thus containing a single pronoun [MASK] to fill. Therefore, as long as a second subject is not introduced in the completion, the probabilistic evaluation will not lose the required context to perform the task correctly.
> >
> > [5] found that the first pronoun generated usually refers to the principal subject, which was the basis for the heuristic to look at only the first pronoun generated when evaluating with TANGO. Therefore, the validity of the constructed templates is likely not a major issue. Nevertheless, we will perform human validation on 100 samples for the generation of a second subject in the next version of this work.
> >
> > -----
> >
> > [5] [“I’m fully who I am”: Towards Centering Transgender and Non-Binary Voices to Measure Biases in Open Language Generation](https://dl.acm.org/doi/10.1145/3593013.3594078) (Ovalle et al., FAccT 2023)

---

> > > ### Comment · Reviewer_U3xB · 2025-06-03
> > > **ACK**
> > >
> > > Thanks for the comprehensive response. It addresses my points. It would be very useful to include this in the main paper (especially the citations) as they ground the motivation for the analysis done and being more specific on recommendations made.

---

> > > > ### Comment · Reviewer_U3xB · 2025-06-03
> > > > **Scores update**
> > > >
> > > > I updated my rating to 7 based on the response.

---

### Official Review · Reviewer_xsTw · 2025-05-11

**Rating:** 8
**Confidence:** 4
**Ethics Flag:** 1

**Summary:**

The paper investigates discrepancies between commonly used methods for measuring misgendering in outputs generated by Large Language Models (LLMs). To conduct this analysis, the authors annotated 2,400 LLM-generated samples with the appropriate gender labels. The results reveal that template-based and probability-based evaluation methods often lead to divergent conclusions, even when applied to the same model architecture. This highlights inconsistencies in current evaluation practices. The paper concludes by offering a set of practical guidelines for selecting appropriate evaluation methods based on specific use cases.

The paper is well written, and the descriptions of the different evaluation methods are both clear and rigorous. The authors state their intention to release the code and dataset, which would represent a valuable contribution to the community and support further research on model evaluation.

**Questions To Authors:**

- What would be your recommendation to evaluate misgendering on a given model?
- Do you think these issues could also be found in other societal bias, such as racial bias?

**Reasons To Accept:**

- The paper is clear and well written. Especially the definition of the different methods and evaluations.
- Misgendering is an important problem for LLMs in tasks like machine translation. Having more rigorous evaluations is worth exploring.

**Reasons To Reject:**

- The paper lacks information about the annotation process and the data used.
- It would be useful to provide more information about possible alternatives to the current misgendering evaluations.

---

> ### Author Response · Authors · 2025-06-02
> **Response to Reviewer xsTw**
>
> Thanks for your thoughtful and helpful feedback! To address your comments:
>
> > “The paper lacks information about the annotation process and the data used.”
>
> We use three publicly-available probability and generation-based evaluation datasets from previous works [1, 2, 3], and we describe how we transform them into parallel versions in Section 4. Furthermore, we describe our annotation methodology in detail in Appendix G. We will release all our data and annotations with the camera-ready version of our paper.
>
> -----
>
> > “It would be useful to provide more information about possible alternatives to the current misgendering evaluations.”, “What would be your recommendation to evaluate misgendering on a given model?”
>
> We recommend using an evaluation format (e.g., probability-based, generation-based) that is more closely aligned with the downstream task (see Section 7). Moreover, we recommend using more naturalistic data, as synthetically-constructed templates and generation contexts may lack ecological validity.
>
> -----
>
> > “Do you think these issues could also be found in other societal bias, such as racial bias?”
>
> Yes, these issues are also relevant to other forms of societal bias. For example, probability-based evaluations may be used to measure the propensity of LLMs to generate stereotypes, which we allude to in lines 62-64. However, such evaluations measure the capacity of LLMs to recognize or reject stereotypes rather than generate them.
>
> [1] [Robust Pronoun Fidelity with English LLMs: Are they Reasoning, Repeating, or Just Biased?](https://aclanthology.org/2024.tacl-1.95/) (Gautam et al., TACL 2024)
>
> [2] [MISGENDERED: Limits of Large Language Models in Understanding Pronouns](https://aclanthology.org/2023.acl-long.293/) (Hossain et al., ACL 2023)
>
> [3] [“I’m fully who I am”: Towards Centering Transgender and Non-Binary Voices to Measure Biases in Open Language Generation](https://dl.acm.org/doi/10.1145/3593013.3594078) (Ovalle et al., FAccT 2023)

---

> > ### Comment · Reviewer_xsTw · 2025-06-05
> >
> > I appreciate that you include the annotation guidelines in section G. Even though more details on section 4 would be useful to better understand the annotation process and your motivation for the decisions you took during the process.

---

### Official Review · Reviewer_fDmf · 2025-05-12

**Rating:** 7
**Confidence:** 4
**Ethics Flag:** 1

**Summary:**

This paper introduces a systematic meta-evaluation of methods across three existing datasets for LLM misgendering. The proposed approach transforms each dataset to enable two types of evaluation: probabilistic and generative. Six LLMs from three different families are evaluated, revealing that these methods do not consistently produce the same results. Disagreements are observed at various levels, including individual instances, datasets, and models, with conflicting results accounting for approximately 20% of all evaluation instances.

In addition to automated metric evaluations, the authors conduct a manual evaluation of 2400 LLM generations, highlighting the complexity of misgendering behaviour, which extends beyond pronouns. The paper concludes with recommendations for future evaluations of LLM misgendering: 1) advocating for evaluation methods that align with the intended deployment, 2) adopting a holistic view of misgendering, 3) recognising the contextual nature of misgendering, and 4) involving individuals most affected by misgendering in the core of system design and evaluation.

**Questions To Authors:**

1. Misgendering is intrinsically linked to gender bias in general. How do you perceive the distinction between misgendering and gender bias in models?
2. To what extent do you believe that the models' difficulty lies in generating the correct gendered pronouns, as opposed to their challenges with coreference resolution and identifying which part of the context should be referred to with a pronoun?

**Reasons To Accept:**

This paper addresses a critical issue in the use of LLMs and NLP more broadly: misgendering. The authors propose a novel methodology that explores both probabilistic and generative approaches to evaluating misgendering in LLMs. They also integrate these two approaches by converting one setup to the other using existing datasets.

The authors investigate the extent of agreement between probabilistic and generative evaluations, demonstrating that there is no single ground truth. They show that models can produce conflicting results depending on how the task is presented, which might not be surprising given the nature of LLMs.

**Reasons To Reject:**

While misgendering is undoubtedly an important issue, I am not convinced that the question of whether generative and probabilistic evaluations yield the same results has not been previously explored. Extensive research in gender bias has demonstrated the fragility of generative outputs, with the same model often producing different results upon multiple runs. Consequently, most studies conduct experiments over several iterations and report the average along with standard deviation values, a practice that is notably absent in this work. HOwever, I do not see this as a severe weakness.

---

> ### Author Response · Authors · 2025-06-02
> **Response to Reviewer fDmf**
>
> Thank you for your feedback and questions!
>
> > “I am not convinced that the question of whether generative and probabilistic evaluations yield the same results has not been previously explored.”
>
> We are not aware of work that explores this question, especially in the context of bias, but would be happy to cite any such work that you know of.
>
> We discuss a lot of parallel work in our related work section (see Section 2) and throughout our paper on: (1) how both probability-based and generation-based evaluations are often brittle (e.g., due to template phrasing, variability of sampling), (2) the correlation between “trick tests” and realistic downstream use cases, and (3) other meta-evaluations (e.g., the correlation between metalinguistic judgments and direct probability measurements).
>
> -----
>
> > “most studies conduct experiments over several iterations and report the average along with standard deviation values, a practice that is notably absent in this work.”
>
> In Figures 3 and 4, we do report the standard deviation of generation-based evaluation results due to sampling variability. These figures suggest that “xe” has low semantic stability, i.e., LLMs fail to correctly produce the neopronoun across different generations for the same context.
>
> However, in Section 5.1, we seek to disentangle the variability of generations due to randomness (i.e., instance-level variation) as an orthogonal issue to the divergence of probability- and generation-based evaluation results due to the variability of evaluation instances (i.e., dataset-level variation). In sum, both the randomness of sampling and brittleness of templates/generation contexts contribute to the disagreement between probability- and generation-based evaluations, but we would like to separate out these dimensions.
>
>
> -----
>
> > “Q: Misgendering is intrinsically linked to gender bias in general. How do you perceive the distinction between misgendering and gender bias in models?”
>
> Misgendering, as it is conceptualized in our and prior work, is most closely related to the testing of models for (binary) gender bias in coreference resolution (e.g., WinoBias [1]). An example instance from WinoBias is: “The physician hired the secretary because [MASK] was overwhelmed with clients.” Evaluations of both misgendering and gender bias in coreference resolution test the capacity of models to non-stereotypically “reason” about social roles (e.g., occupations). However, evaluations of misgendering, unlike traditional evaluations of gender bias in coreference resolution, also assess the capacity of models to use pronouns correctly in context and consider singular “they” and neopronouns.
>
> -----
>
> > “To what extent do you believe that the models' difficulty lies in generating the correct gendered pronouns, as opposed to their challenges with coreference resolution and identifying which part of the context should be referred to with a pronoun?”
>
> Both of these aspects contribute to misgendering. [2] seeks to disentangle reasoning about coreference resolution from shallow repetition of pronouns and reverting to biased predictions in the context of “he,” “she,” “they,” and “xe.” We believe that for neopronouns like “xe,” model exhibit general difficulty in generating the correct pronoun, whereas models have no issue generating “he” and “she” but are more easily thrown off by non-stereotype-conforming pairings:
> - “The physician hired the secretary because *she* was overwhelmed with clients.” [1]
> - “Elizabeth’s pronouns are he/him/his. Elizabeth published a book. Please go to…”  (see Table 9 in the appendix)
>
> -----
>
> [1] [Gender Bias in Coreference Resolution: Evaluation and Debiasing Methods](https://aclanthology.org/N18-2003/) (Zhao et al., NAACL 2018)
>
> [2] [Robust Pronoun Fidelity with English LLMs: Are they Reasoning, Repeating, or Just Biased?](https://aclanthology.org/2024.tacl-1.95/) (Gautam et al., TACL 2024)

---

> > ### Comment · Reviewer_fDmf · 2025-06-10
> >
> > Thank you for your detailed response. You addressed my points and clarified what I asked about. This reinforces my initial impression of the paper’s overall quality.

---

### Author Response · Authors · 2025-06-02

Thanks to all the reviewers for their recognition of the relevance of our work and positive feedback:
- Reviewer Zan3 said “the generation of parallel datasets for probabilistic and generative evaluations is novel” and “the experimental part is solid.”
- Reviewer U3xB found that “the analysis from human evaluations are insightful.”
- Reviewer xsTw commented that “the paper is clear and well written.”
- Reviewer fDmf felt that “the authors propose a novel methodology that explores both probabilistic and generative approaches to evaluating misgendering in LLMs.”

We address each of the reviewer’s comments separately below.

---

### Decision · Program_Chairs · 2025-07-08

**Decision:**

Accept

**Comment:**

This paper focuses on LLM misgendering (for example, getting someone’s pronoun wrong). It considers 3 misgendering datasets, converts them into probabilistic (cloze-style) and generative (sampling) versions, and then shows for six models that the methods can disagree at the instance, dataset and model level. The paper finally also provides a human evaluation that sheds light on the complexity of the problem of misgendering.

Reasons to accept: Well written (xsTw), important topic (fDmf, xsTw, Zan3), explores probabilistic and generative approaches (fDmf, Zan3),  multiple models and datasets (U3xB), useful human evaluation (U3xB, Zan3)

Reasons to reject: Lack of correlation between the two evaluation types is not inherently problematic; it highlights the different capacities (U3xB), human eval was done by the authors themselves which is not ideal (Zan3), the recommendations are (overly) general (U3xB), whether generative and probabilistic evaluations yield the same results previously explored (fDmf), paper does not report/explore averaged results from multiple runs with standard deviations (fDmf), needs more info on annotation process (xsTw), would be useful to provide more information about possible alternatives to the current misgendering evaluations. (xsTw), the evaluation part is dense with a lot of results and discussion. (Zan3)

Overall the reasons to accept by far outweigh the reasons to reject, and this is reflected by the reviewers' scores who agree on a recommendation to accept the paper. Everyone agrees this is an important topic, and the comparison between generative and probabilistic evaluation is well done along multiple axes (datasets, models), even though the authors indeed agree with one of the reviewers that one should not expect these methods to align, and the paper should make this clear. Some other concerns were alleviated during discussion, for example details on the evaluation process. Ideally the human evaluation would not have been done by the authors, but it's not a reason to reject the paper given its contributions. The eval sect could be improved. Finally, the authors might find the following references useful for the next revision of the paper: [1] on converting probabilistic into generative evals, and [2] on misgendering in machine translation and LLMs.

[1] Escaping the sentence-level paradigm in machine translation. Post and Junczys-Dowmunt.
[2] MiTTenS: A Dataset for Evaluating Gender Mistranslation. Kevin Robinson et al.